# Adsorption of Mono- and Divalent Ions onto Dendritic Polyglycerol Sulfate (dPGS) as Studied Using Isothermal Titration Calorimetry

**DOI:** 10.3390/polym15132792

**Published:** 2023-06-23

**Authors:** Jacek J. Walkowiak, Rohit Nikam, Matthias Ballauff

**Affiliations:** 1DWI—Leibniz-Institute for Interactive Materials e.V, Forckenbeckstraße 50, 52074 Aachen, Germany; 2Institute of Technical and Macromolecular Chemistry, RWTH Aachen University, Worringerweg 2, 52074 Aachen, Germany; 3Aachen-Maastricht Institute for Biobased Materials (AMIBM), Maastricht University, Urmonderbaan 22, 6167 RD Geleen, The Netherlands; 4Helmholtz-Zentrum Berlin für Materialien und Energie, Hahn-Meitner-Platz 1, 14109 Berlin, Germany; rohit.nikam@helmholtz-berlin.de; 5Institut für Chemie und Biochemie, Freie Universität Berlin, Taktstraße 3, 14195 Berlin, Germany; mballauff@zedat.fu-berlin.de

**Keywords:** isothermal titration calorimetry (ITC), counterion condensation, dendritic polyglycerol sulfate (dPGS)

## Abstract

The effective charge of highly charged polyelectrolytes is significantly lowered by a condensation of counterions. This effect is more pronounced for divalent ions. Here we present a study of the counterion condensation to dendritic polyglycerol sulfate (dPGS) that consists of a hydrophilic dendritic scaffold onto which sulfate groups are appended. The interactions between the dPGS and divalent ions (Mg^2+^ and Ca^2+^) were analyzed using isothermal titration calorimetry (ITC) and showed no ion specificity upon binding, but clear competition between the monovalent and divalent ions. Our findings, in line with the latest theoretical studies, demonstrate that a large fraction of the monovalent ions is sequentially replaced with the divalent ions.

## 1. Introduction

Polyelectrolytes (PEs) are polymers bearing charged groups with dissociable counterions. Thus, a single PE may have a nominal charge on the order of 10^6^ e. However, in solution, a part of the PE’s nominal charge is balanced by counterion condensation. In that way, a fraction of closely correlated counterions determines a much smaller effective charge of a given PE [1,2]. For rod-like macroions, this effect was described many decades ago by Manning [3]. Since that time, the concept has been enlarged to comprise PEs with different architectures, such as polyelectrolyte brushes [4,5,6,7], dendritic polyelectrolytes [8], or charged gels [9,10]. In 2019, Staňo and co-workers investigated the effect of multivalency with respect to ionization and conformational changes of weak, star-like polyelectrolytes [11]. They showed that in a mixture of monovalent and multivalent ions, the multivalent ions are found almost exclusively in the star interior, whereas the monovalent ions remain in the solution. The branching point of such polyelectrolytes is characterized by the highest concentration of ionizable monomers, which results in the accumulation of multivalent counterions.

The response of PEs to their environment has been extensively studied, as it can help with understanding the regulation of ion transport [12], interactions with surfaces (adhesion, wettability, lubrication) [13,14], and the response to external triggers (optoelectronics) [15].

In the approach described by Plamper et al., the structure of the star-shaped PEs was studied in the presence of tri- and divalent ions [16]. Using photochemical reactions, trivalent counterions were converted into a mixture of mono- and divalent ions, thus switching the star PEs from the collapsed to the expanded state. The accumulation of counterions determined by their valency is virtually quantitative [17], and the effect of multivalent salts on PE brushes has been systematically studied in recent years [18,19,20,21,22]. In 2018, Yu et al. reported that the excellent lubrication properties of a PE brush in a solution of monovalent counterions can be diminished upon introduction of multivalent ions [23]. Moreover, charge regulation plays a crucial role in PE–protein interactions, having a direct impact on the applicability and working conditions of biomedical devices [24,25].

Recently, dendritic polyglycerol sulfate (dPGS) has become a much-studied PE due to its potential use in various medical applications [26,27] and as an alternative to heparin. dPGS consists of a highly hydrophilic scaffold made of poly(glycerol) onto which sulfate groups are appended. The binding of dPGS to proteins in aqueous solution has been studied in detail using isothermal titration calorimetry (ITC) [28,29,30]; a survey of these studies was reviewed recently [31]. These studies have clearly revealed the importance of the condensed counterions to the formation of complexes with proteins: upon the binding of a protein to dPGS, some of the condensed counterions are released into the bulk phase. The concomitant gain of free energy is the main driving force for binding [28,31].

Up to now, all experimental studies using dPGS [31] have been performed for monovalent counterions. Molecular dynamics (MD) simulations revealed a strong correlation of the counterions with the dendritic macroion [8,32], which led to a marked decrease in the effective charge of the macroion also seen in experiments [8]. Divalent ions such as Mg^2+^ or Ca^2+^ play an important role in biochemical processes and will certainly be highly correlated with macroions in solution. Recently, a combination of coarse-grained simulations with analytical models was used to quantify the competition between mono- and divalent ions in binding to a highly charged macroion [33,34]. The authors demonstrated that the monovalent counterions condensed on the dPGS dendrimer were sequentially replaced with divalent ions, leading to a significant decrease in the effective charge and the effective potential of the dendrimer. As a consequence, the electrostatic attraction between the dPGS and human serum albumin (HSA) was effectively reduced at a high concentration of divalent ions. In particular, MD simulations led to a well-defined ratio between bond divalent and monovalent counterions as a function of the divalent ions’ concentration. Hence, the MD simulation results can be quantitatively compared with the experiment.

In the present paper, we study the interaction of dPGS with divalent ions in aqueous mixtures of mono- and divalent ions. We show that ITC can be used to monitor the replacement of condensed monovalent counterions by divalent magnesium or calcium ions. The results of the present study can be directly compared with the data provided by MD simulations [34]. Figure 1 shows this process in a schematic fashion: we start from a solution of a second-generation dPGS (dPGS-G2) [8] having monovalent counterions, and add stepwise a solution containing divalent ions. The exchange of the monovalent and divalent ions is accompanied by a small but measurable heat effect that can be detected with ITC. Hence, the competitive interaction of ions differing in valency with a macroion can be studied with ITC in the same way as established already for the complex formation of macroions with proteins [31]. The experimental results thus obtained can then be compared with analytical models developed recently [33,34].

## 2. Materials and Methods

### 2.1. Materials

The 3-(N-morpholino)propane sulfonic acid (MOPS, 99.5%, Aldrich, Darmstadt, Germany), sodium chloride (NaCl, ≥99.0%, Aldrich, Darmstadt, Germany), calcium chloride dihydrate (CaCl_2_ ∙ 2H_2_O, ≥99.0%, Aldrich, Darmstadt, Germany), and magnesium chloride hexahydrate (MgCl_2_ ∙ 6H_2_O, 99.0%, Aldrich, Darmstadt, Germany) were used without further purification. The water was purified by filtration thorough a millipore system (Merck, Darmstadt, Germany) resulting in resistivity higher than 18 MΩcm.

The dendritic polyglycerol sulfate (dPGS) was obtained by sulfation of a fractionated hyperbranched polyglycerol [35] and kindly provided by AG Haag from the Institut für Chemie und Biochemie at Freie Universität Berlin. Figure 2 displays the chemical structure of dPGS. Table 1 gives the molecular weight M_n,dPGS_ of dPGS. The degree of sulfation (DS) was determined from the weight percentage of sulfur [36].

The properties of dPGS gathered in Table 1 show that the number of terminal sulfate groups N_ter_ is 34. That differs from a perfect chemical structure of dPGS-G2, in which approximately 24 terminal sulfate groups are present [29].

### 2.2. Isothermal Titration Calorimetry

ITC experiments were conducted on a Microcal VP-ITC instrument (Microcal, Northampton, MA, USA). All samples used in the measurements were prepared in a water-buffer solution of 10 mM MOPS and such a NaCl concentration to adjust to a certain ionic strength after the final injection of the titrant (Mg^2+^, Ca^2+^). The pH of each solution was fixed to 7.2. A total of 280 µL of Mg^2+^/Ca^2+^ buffer solution was titrated with 35 successive injections of 8 μL each into the cell containing 1.43 mL of dPGS solution. The stirring rate of 307 rpm was set with a time interval of 300 s between each injection. The concentrations of divalent ions in the injectant and of dPGS are listed in Table 2. The measurements were performed at 30 °C. Before each experiment, all samples were degassed and thermostatted for several minutes at 1 degree below the experimental temperature.

The evaluation of the ITC data is demonstrated in Figure 3, which shows the raw ITC signal of the binding (black curves and squares) and the dilution of the divalent-ion solution (red curves and squares). For further analysis, the heat of dilution of the divalent ions was subtracted from the heat of adsorption.

### 2.3. Data Analysis

#### Single Set of Identical Binding Sites (SSIS) Model

The SSIS model, based on the Langmuir equation, [37] assumes an equilibrium between the unoccupied binding sites of dPGS, the number of divalent ions in solution, and the occupied binding sites. It relates the fraction of the adsorption sites in the dPGS containing bound divalent ions *θ* to the binding constant *K_b_*:(1)$$\theta =\frac{K_{b}\left[A\right]}{1+K_{b}\left[A\right]}$$
where [*A*] is the concentration of free divalent ions in solution. Because the total concentration of [*A*]*_tot_* in the solution is known, [*A*] is connected to [*A*]*_tot_* as follows:(2)$${\left[A\right]}_{tot}=\left[A\right]+N\theta \left[dPGS\right]$$

For dPGS containing *N* adsorption sites, *θ* is *N_b_/N*, where *N_b_* represents the number of divalent ions bound per dPGS molecule and [*dPGS*] is the total concentration of dPGS in solution. Details on solving Equation (1) for *θ* can be found in the Appendix A.

The heat *Q′* after each injection *i* within the volume *V*_0_ of the calorimetric cell is equal to:(3)$$Q^{\prime }=\left[dPGS\right]V_{0}N\theta ∆H^{ITC}$$

The experimental data are fitted by calculating the heat change in the solution ∆*Q_i_* released with each injection *i* and corrected for displaced volume ∆*V*_i_:(4)$$∆{Q^{\prime }}_{i}={Q^{\prime }}_{i}+\frac{dV_{i}}{V_{0}}\left[\frac{Q_{i}+Q_{i-1}}{2}\right]-{Q^{\prime }}_{i-1}$$

Fitting of the experimental data involves initial guesses for *N_b_*, *K_b_*, and ∆*H^ITC^*; the calculation of ∆*Q′_i_* for each injection and a comparison of these values with the measured heat for the corresponding experimental injections; the improvement in the initial values on the basis of the Marquardt methods; and the iteration of the above procedure until a satisfactory fit is achieved [38].

## 3. Results and Discussion

After evaluation of the ITC data described in Section 2.2., the integrated isotherms were fitted with the single set of identical binding sites (SSIS) model (see Section 2.3.) The thermodynamic parameters for the binding are listed in Table 3 (highlighted in blue). All signals of the binding of divalent ions to dPGS were endothermic in the entire range, indicating that the driving force of this process is of entropic origin [39].

### 3.1. Ion Specificity

Possible effects of ion specificity on the interactions of divalent ions can be analyzed by studying the adsorption of Mg^2+^ and Ca^2+^ to dPGS. The ITC isotherms fitted with the SSIS model are presented on a semi-logarithmic plot in Figure 4. The respective parameters (see Table 3) show that in the limit of error, there is no difference in the binding of Mg^2+^ and Ca^2+^ to dPGS. A similar observation with respect to PE brushes was reported recently by Xu et al.; the authors systematically studied the ion-specific effects of the divalent cations Mg^2+^, Ca^2+^, and Ba^2+^ on the structure of polystyrene sulfonate (PSS) brushes [40]. The reported ITC results showed no significant difference in the binding energy between Mg^2+^ and Ca^2+^. Although the binding energy between Ba^2+^ and PSS was much higher than that of Mg^2+^ and Ca^2+^, the difference between the latter two was marginal. Thus, the effective charge of dPGS should depend only on the valency of the condensed counterions, and the influence of their size (with regard to Mg^2+^ and Ca^2+^) can be neglected in the first approximation. Based on that, the further analysis is focused on the interaction of dPGS with Mg^2+^ ions.

### 3.2. Constant Ionic Strength and Increasing Concentration of Mg^2+^

The ITC isotherms for three different concentrations of Mg^2+^ ions are presented in Figure 5; the diagrams containing the raw data are gathered in the Appendix A. The resulting thermodynamic data are listed in Table 3 (highlighted in tan).

The ITC curve profiles shown in Figure 5 indicate nonspecific binding [41], where an increased concentration of Mg^2+^ ions leads to a greater thermal effect during the titration experiment. The resulting binding parameters (see Table 3) show that at a constant ionic strength of 21.5 mM, the number of divalent ions adsorbed to dPGS increases with their increasing concentration. Thus, the number of bound Mg^2+^ ions increases as a function of the decreasing concentration of Na^+^. This trend highlights the competition between divalent and monovalent ions upon adsorption to dPGS. This result stands in agreement with recent theoretical work on the charge renormalization effect on dPGS in solution with mono- and divalent ions [34]. The authors adopted a binding model based on the Donnan equilibrium that gives insight into ion condensation and compared it with molecular dynamics (MD) simulations. They showed that the monovalent cations condensed on the dPGS are sequentially replaced with divalent ions due to the stronger electrostatic attraction between the divalent ions and the dendrimer. This leads to a decreased effective charge and effective potential of the dendrimer, affecting its binding pathway with HSA. Because the driving force of the studied interaction was of entropic origin due to the release of monovalent counterions, the complexation of dPGS with divalent ions led to a comparatively smaller number of released counterions and a lower entropic effect. Therefore, charge regulation of dPGS by mono- and divalent ions, combined with insights into the effect of counterion release, provides a key understanding of the activity of protein inhibitors in general.

Figure 6 shows the ratio between the divalent and monovalent cations Nb,++/Nb,+ bound to dPGS as a function of the divalent ion concentration, c++. The MD simulation and the Donnan modeling data are taken from Figures 5 and 6 of ref. [34]. The Nb,++/Nb,+ ratio between the divalent and monovalent cations bound to dPGS shows no significant difference from the calorimetric data up to a Mg^2+^ concentration of 1 mM. For higher concentrations, ITC shows approximately half as many bound divalent ions as the simulation (cf. Figures 5 and 6: dPGS-G2 in ref. [34]). The deviation between the Donnan model and the simulation is due to the strong charge–charge correlation, which is beyond the reach of the former. In turn, the smaller number of condensed Mg^2+^ ions measured with ITC may be a consequence of the weak complexation of metal ions with the MOPS buffer [42,43,44]. Although widely used for pH control in biological and biochemical research, it can lead to a notable decrease in condensed Mg^2+^ ions compared with the simulation, which did not include the buffer presence. Certainly, several other factors might contribute to this effect, e.g., structural imperfections of the dPGS molecules and their deviation from a perfect dendrimer, as well as the ions and polymer solvation effects that the simulation is not able to address [45,46,47]. Such a discrepancy is naturally of great importance, as the authors demonstrated in a theoretical study of the complexation between dPGS and HSA at high concentrations of divalent ions [34]. Due to the decreased effective charge and the effective potential of dPGS at high concentrations of divalent ions, the electrostatic attraction between the dendrimer and HSA was effectively reduced.

It is worth emphasizing that measurements performed at low ionic strength (16.5–21.5 mM) resulted in well-defined ITC isotherms with a sufficient number of data points, as presented in Figure 4 and Figure 5. This excludes the possible underestimation of the number of bound Mg^2+^ ions as a result of an inaccurate SSIS model fit due to an undefined inflection point. The latter is clearly visible on a semi-logarithmic plot in Appendix A. In contrast to that, the measurement performed at I = 31.5 mM led to a dramatic decrease in the measured enthalpy (see Appendix A), making it impossible to fit the SSIS model effectively. This indicates that the studied process, although it allows us to analyze the competitive binding between monovalent and divalent ions to the target dPGS molecule, is, at the same time, extremely sensitive to an increase in ionic strength. Because the latter lowers the binding affinity, it makes a sufficiently precise analysis impossible at high concentrations of counterions.

## 4. Conclusions

This paper presents calorimetric measurements of the interaction of dendritic polyglycerol sulfate (dPGS) with divalent and monovalent ions in aqueous solution. It shows that the ion-specific effects are marginal for Mg^2+^ and Ca^2+^ ions upon binding to dPGS but reveals a clear competition between mono- and divalent ions with regard to that process. It shows that at a constant ionic strength of I = 21.5 mM, the fraction of adsorption sites in dPGS containing bound Mg^2+^ ions increases with their increasing concentration. This means that the number of bound Mg^2+^ ions (Nb,++) is inversely proportional to the concentration of Na^+^. The quantitative description of the competitive binding between mono- and divalent ions and dPGS through isothermal titration calorimetry (ITC) provides a crucial experimental basis for further studies on dPGS and its complexation with proteins. Finally, it shows that despite the small heat effect, ITC is well suited to monitor the discussed ion exchange.

## Figures and Tables

**Figure 1 polymers-15-02792-f001:**
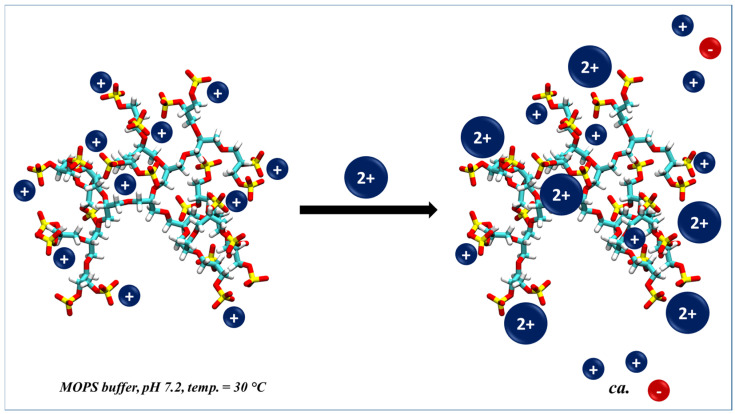
Schematic illustration of the competitive binding between divalent and monovalent cations to dPGS in water medium. Red and blue colors indicate the negative and positive charge, respectively.

**Figure 2 polymers-15-02792-f002:**
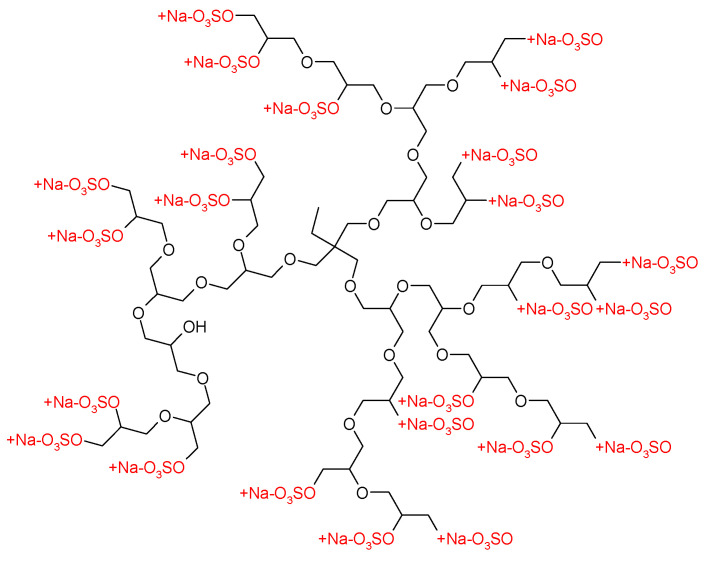
Chemical structure of a perfect dPGS-G2 dendrimer.

**Figure 3 polymers-15-02792-f003:**
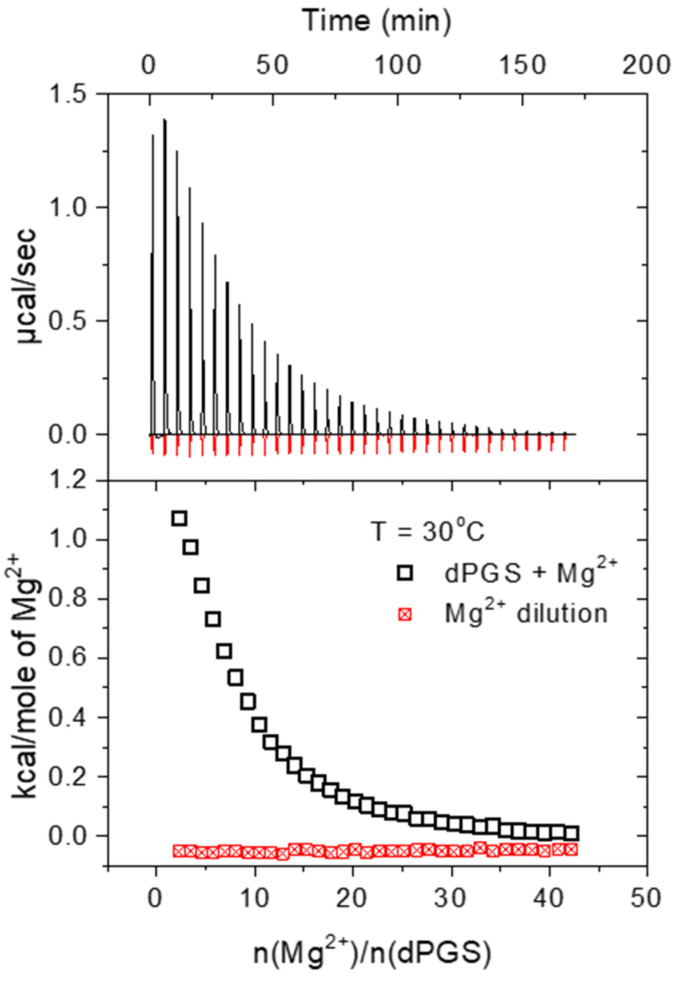
ITC data for the binding of Mg^2+^ ions to dPGS at pH 7.2 and temperature of 30 °C in 10 mM MOPS buffer. The upper panel shows the raw data of the binding (black spikes) and the dilution of Mg^2+^ by the buffer (red spikes). The integrated heats of each injection are shown in the lower panel.

**Figure 4 polymers-15-02792-f004:**
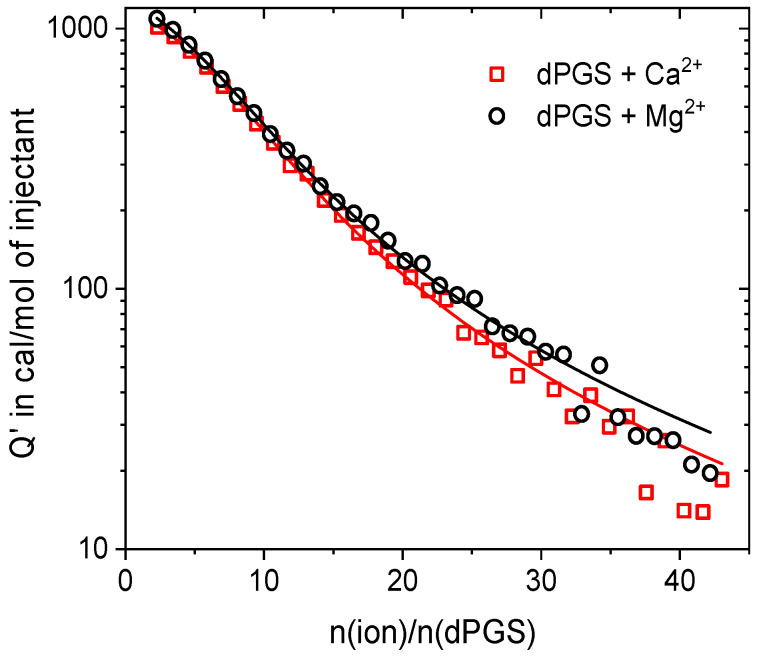
Binding isotherms for Ca^2+^ and Mg^2+^ interacting with dPGS. Solid lines represent the SSIS fit. Thermodynamic data resulting from the fitting are listed in Table 3 (highlighted in blue).

**Figure 5 polymers-15-02792-f005:**
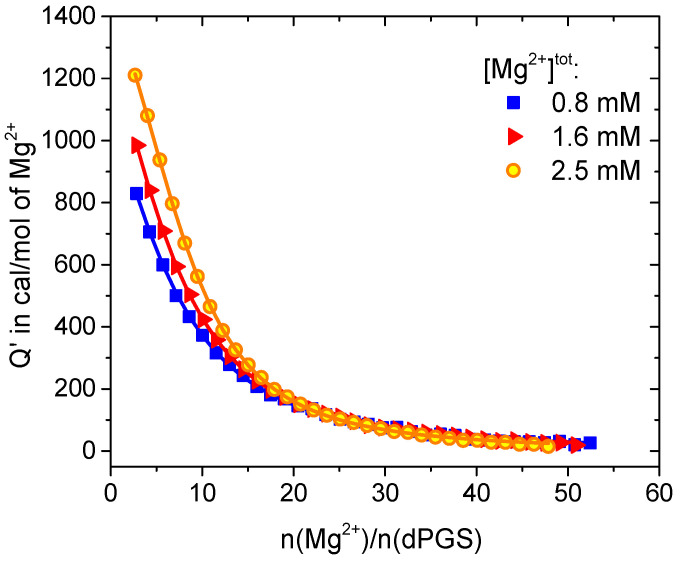
Binding isotherms for Mg^2+^ interacting with dPGS. Solid lines represent the SSIS fit. Thermodynamic data resulting from the fitting are listed in Table 3 (highlighted in tan).

**Figure 6 polymers-15-02792-f006:**
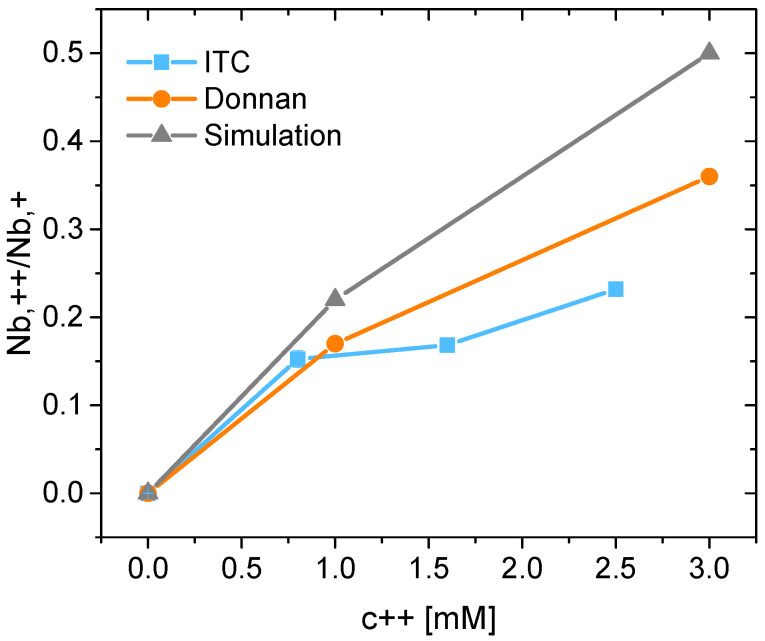
The ratio of the divalent and monovalent cations Nb,++/Nb,+ bound to dPGS as a function of the divalent ions concentration, c++. Solid lines are used for eye guidance. Results from the simulation and the Donnan modeling are taken from Figures 5 and 6 of ref. [34].

**Table 1 polymers-15-02792-t001:** Properties of dPGS.

	dPGS
M_n,dPG_ (kD)	2.6
DS (%)	97
N_ter_	34
M_n,dPGS_ (kD)	6.5

DS: the degree of sulfation determined from elemental analysis. N_ter_: the number of terminal sulfate groups. The number-average molecular weight M_n_ of the dPG core, as well as that for dPGS, was determined using gel permeation chromatography.

**Table 2 polymers-15-02792-t002:** Experimental parameters for dPGS–divalent ion measurements conducted on a VP-ITC instrument.

System	Buffer/Ionic Strength (mM)	[DI]^tot (a)^ (mM)	[Na^+^]^tot^ (mM)	T [K]	c (DI) ^(b)^ (mM)	c (dPGS) (mM)
Ca^2+^/dPGS	MOPS/16.5	0.8	4.1	303	5.1	0.032
Mg^2+^/dPGS	MOPS/16.5	0.8	4.1	303	5.0	0.032
	MOPS/21.5	0.8	9.1	303	5.0	0.020
	MOPS/21.5	1.7	6.4	303	10.0	0.039
	MOPS/21.5	2.5	4.0	303	15.2	0.064

^(a)^ DI—divalent ion; ^(b)^ concentration of divalent ions in the injectant.

**Table 3 polymers-15-02792-t003:** Thermodynamic parameters for the binding of divalent ions to dPGS resulting from the SSIS fit.

Divalent ion (DI)	[DI]^tot^ (mM)	[Na^+^]^tot^ (mM)	[dPGS] (mM)	I (mM)	N_b_	∆H^ITC^ (kJ∙mol^−1^)	K_b_ × 10^−3^ (M^−1^)	∆G_b_^exp^ (kJ∙mol^−1^)
Ca^2+^	0.8	4.1	0.032	16.5	7.9 ± 0.2	6.8 ± 0.2	6.3 ± 0.5	−22.0 ± 0.2
Mg^2+^	0.8	4.1	0.032	16.5	7.5 ± 0.2	8.1 ± 0.4	5.1 ± 0.4	−21.5 ± 0.2
	0.8	9.1	0.020	21.5	4.5 ± 0.3	10.1 ± 0.8	6.5 ± 0.6	−22.1 ± 0.2
	1.6	6.4	0.039	21.5	4.9 ± 0.1	10.0 ± 0.3	4.1 ± 0.2	−21.0 ± 0.1
	2.5	4.0	0.064	21.5	6.4 ± 0.1	8.5 ± 0.1	4.1 ± 0.2	−21.0 ± 0.1

Data highlighted in blue are discussed in Section 3.1.; data highlighted in tan are discussed in Section 3.2. N_b_ represents the number of divalent ions bound per dPGS (Equation (2)); K_b_ is the binding constant (Equation (1)); and ∆H^ITC^ is the calorimetric heat (Equation (3)). The Gibbs free energy of binding (ΔG_b_^exp^) is given by ΔG_b_^exp^ $=-RTlnK_{b}$, where *R* is the gas constant and *T* is the absolute temperature.

## Data Availability

The data presented in this study are available within the article and Appendix A.

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
