# Peer review of "Adsorption of Mono- and Divalent Ions onto Dendritic Polyglycerol Sulfate (dPGS) as Studied Using Isothermal Titration Calorimetry"

_polymers, 2023, doi:10.3390/polym15132792_

Round 1

Reviewer 1 Report

This is an excellent contribution relating to the partitioning of higher valent ions to model branched polyelectrolytes. The measurements are very interesting and be should helpful in helping to improving the theory of these complex solutions.

The only shortcoming of the paper relates to the absence of any discussion of the importance of ion and polymer hydration, which leads to manifold specific ion effects such as the Hoffmeister series, chaotropic and kosmotropic. Even the existence of kosmotropic ions cannot be explained by models that treat the solvent as a continuum or even by conventional class-cal molecular dynamics simulations with explicit water.

The authors study a phenomenon that is evidently not sensitive to specific ion effects in relation to simulations not capable of addressing and they find reasonable agreement. This fine and interesting, but there is evidence that ion hydration can play an important role in ion partitioning in polymers, as evidenced by Collins empirical "law of matching affinities", which not comprehensible in general from modeling treatments that neglect ion and polymer hydration.

I only request that the authors add a statement indicating that the modeling they cire does no address ion and polymer solvation effects that might become important in some systems,

Otherwise, I find the paper perfectly acceptable for publication in its present form.

Reviewer 2 Report

Authors studied counterion condensation on strongly charged polyelectrolytes.

When both divalent and monovalent ions are present, adsorption of divalent ions is advantageous because twice as much monovalent ions can be released to obtain large entropy.

Authors quantitatively obtained the amount of adsorbed divalent ions using ITC method and tried to determine the adsorption ratio of divalent and monovalent ions by comparing experimental data with simulation data and theory.

The discrepancy between simulation or simple theory and the current data is not quite satisfactory (Fig.6) to support experimental findings.  It would be beneficial for the readers to provide better explanation for such discrepancies.   

In line 243, authors attributed smaller number of Mg2+ ions in ITC experiments to weak complexation of metal ions with the MOPS buffer. 

Would using different buffer improve it?

Does the simulation (Ref.34) include buffer condition? Are the simulation condition and experimental condition equivalent including dPGS concentration?

I wonder why the authors use the SSIS model for analysis. If charge-charge correlation is important, monovalent ions should enter the chemical equation to determine the equilibrium concentration of condensed divalent ions.   

Please be more precise on following statements.

The nominal charge of PE can range from highly negative to highly positive, depending on the specific polymer and pH of the solution. 

What does mean by PE has nominal charge of 106

Reviewer 3 Report

This paper demonstrates through detailed experimental results that the dendritic polyglycerol sulfate (dPGS) undergoes a transition from monovalent to divalent cations, which can be observed using isothermal titration calorimetry (ITC). However, the paper primarily focuses on describing the experimental findings and lacks a thorough explanation of why the monovalent cations are replaced by divalent cations. After addressing this point and providing a detailed explanation, it would be recommended to consider publishing the paper.

Round 2

Reviewer 2 Report

The revised manuscript can be published. 

Reviewer 3 Report

I have reviewed the points raised and confirmed that they are adequately explained. This paper is recommended for publication.